# Exploration of the Impact of Cybersecurity Awareness on Small and Medium Enterprises (SMEs) in Wales Using Intelligent Software to Combat Cybercrime

**Nisha Rawindaran** [1,2]**, Ambikesh Jayal** [3,*] **and Edmond Prakash** [4]

1　Cardiff School of Technologies, Cardiff Metropolitan University, Cardiff CF5 2XJ, Wales, UK
2　Aytel Systems Ltd., Cardiff CF3 2PU, Wales, UK
3　School of Information Systems and Technology, University of Canberra, Bruce, ACT 2617, Australia
4　Research Centre for Creative Technologies, University of the Creative Arts, Farnham GU9 7DS, Surrey, UK
*　Correspondence: ambi.jayal@canberra.edu.au

**Abstract:** Intelligent software packages have become fast-growing in popularity for large businesses in both developed and developing countries, due to their higher availability in detecting and preventing cybercrime. However, small and medium enterprises (SMEs) are showing prominent gaps in this adoption due to their level of awareness and knowledge towards cyber security and the security mindset. This is due to their priority of running their businesses over requiring using the right technology in protecting their data. This study explored how SMEs in Wales are handling cybercrime and managing their daily online activities the best they can, in keeping their data safe in tackling cyber threats. The sample collected consisted of 122 Welsh SME respondents in a collection of data through a survey questionnaire. The results and findings showed that there were large gaps in the awareness and knowledge of using intelligent software, in particular the uses of machine learning integration within their technology to track and combat complex cybercrime that perhaps would have been missed by standard cyber security software packages. The study's findings showed that only 30% of the sampled SMEs understood the terminology of cyber security. The awareness of machine learning and its algorithms was also questioned in the implementation of their cyber security software packages. The study further highlighted that Welsh SMEs were unaware of what this software could do to protect their data. The findings in this paper also showed that various elements such as education and the size of SME made an impact on their choices for the right software packages being implemented, compared to elements such as age, gender, role and being a decision maker, having no impact on these choices. The study finally shares the investigations of various SME strategies to help understand the risks, and to be able to plan for future contingencies and preparation in keeping data safe and secure for the future.

**Keywords:** machine learning; cyber security; SME; intelligent software; cyber threats

## 1. Introduction

Wales is a leading hub in the United Kingdom's fast-growing cyber security sector. Small- and medium-sized enterprises (SMEs) in Wales are changing in its technology landscape. Wales is the home for thriving businesses contributing back to the Welsh economy. With an estimated 262,800 enterprises, both large and small, that are active in Wales, employing an estimated 1.2 million people, SMEs in Wales account for 62.6% of employment and 39.4% of turnover, with large enterprises accounting for the remainder in 2021. Therefore, the majority of active enterprises of SMEs (0–249 employees), accounted for 99.4% of total enterprises in Wales in 2021. Micro enterprises (0–9 employees) accounted for 95.0% of the total enterprises in Wales. In the same year, wholesale, retail, transport, hotels, food and communication were the largest sectors in Wales, with 65,200 enterprises and the employment of 394,300 people. Being a devolved nation of the United Kingdom, for Wales,

the effects of transfer of power from central government to local or regional administration has allowed decisions to be made independently, especially for digital transformation [1].

The digital strategy for Wales was first released in 2021, with the purpose of this strategy to take a look ahead and set out a national vision for jointly adopting a digital approach across Wales [2]. When it first was released, it was to ensure the people in Wales experienced modern and efficient living through its public services supported by good, ethical use of data. It was also to stimulate innovation in our economy and support businesses to develop the resilience Wales needs to succeed. Digital confidence was also aforementioned in engaging communities and business in modern society. The importance of all learners to acquire knowledge, experience and skills further benefits from an increasingly digital and changing economy. Since the pandemic, the rise in SMEs using the internet to operate online has increased, this of course magnifying the threat landscape, creating greater vulnerabilities and exposure to attacks. With the recent amendments to the General Data Protection Regulation (GDPR) Laws in the UK in 2018 [3], Wales as a nation has had to cope with these changes and help SMEs roll out the data protection program and change the way data is managed and stored. The Welsh government has strengthened its initiatives and introduced targets for businesses to achieve.

This paper will explore SME participants through a survey questionnaire, all of them representing SME businesses located in Wales. The study aimed to apply convenience sampling on the data collected through a methodology of positivism using a quantitative approach, to make sense of the data that have been collected. Results will be analyzed to identify patterns and trends in the data collected. The study will explore Welsh SME cybersecurity awareness, knowledge of cybersecurity, and how they contribute to SMEs' resistance or willingness to move forward with this concept. It will also highlight how engagement with cybersecurity agencies can boost SMEs' digital confidence in tackling complex problems. SMEs need to be aware that cyberattacks will eventually hit their business and that it is only a matter of time, and if they are not prepared for this process, the loss will be detrimental to the SME business. Getting SMEs to understand this concept is indeed a challenge in itself and can only get better if digital maturity is in place. The remainder of the paper is organized as follows: Section 2 provides background and theoretical context. Section 3 reviews related work and previous literatures upon which this research builds. Section 4 addresses the methodology and design of survey. Section 5 contains the results and analysis. Section 6 reviews the limitations to the research. The concluding section contains the conclusion and future work.

## 2. Background and Theoretical Context

### 2.1. Governance

Wales follows the cyber security laws such as the GDPR. The Data Protection Act (DPA) (2018) and the UK GDPR is the law in relation to personal data. The GDPR is used along with the EU General Data Protection Regulation which, following the UK's withdrawal from the EU, has been incorporated into the laws of England and Wales, Scotland and Northern Ireland by virtue of the European Union (Withdrawal) Act 2018 and as amended by the Data Protection, Privacy and Electronic Communications (Amendments etc.) (EU Exit) Regulations 2019 (SI 2019/419)) ("UK GDPR") [4]. Wales also follows the Privacy and Electronic Communications (EC Directive) Regulations 2003 ("PECR") that sits alongside the DPA 2018 and the UK GDPR, providing specific privacy rights in relation to electronic communications. The Computer Misuse Act (CMA) 1990 legislation covers various cybercrime offences aiming to secure computer material against unauthorized access or modification. Lastly, the Network and Information Systems Regulations 2018 ("NIS Regulations"), which the UK Government are currently looking to update, provide legal measures to increase the level of security (both from a cyber-perspective and in terms of physical resilience) of networks and information systems for the provision of essential services (for example water, transport, energy, healthcare and digital infrastructure) and digital services (search engines, online marketplace or cloud computing services). With

such laws in place for a devolved nation of Wales, it is imperative that SMEs follow suit and are protected under this governance; however, it remains to be seen regarding an attack occurring in an SME business and how protected they are. Under the Welsh Procurement Policy Note WPPN 08/21: Cyber Essentials, wherein, in a country-wide effort to reduce the levels of cyber security risk in the supply chain, the UK Government via the National Cyber Security Centre (NCSC) developed the Cyber Essentials Scheme to ensure a minimum level of security for all their suppliers and businesses alike, who must adhere to the scheme in order to comply. For Wales, Cyber Essentials is required for all relevant Welsh Government contracts and is used by the Welsh Government. Both the Cyber Essentials and Cyber Essentials Plus is the highest level of certification offered under this Cyber Essentials scheme, which is an official UK-wide, government-backed certification. Whilst some government contracts may require you to be Cyber Essentials-certified or to be able to demonstrate that the technical controls are in place, it is not mandatory for all businesses to comply, hence leaving the responsibility to the SME business to decide the importance of cyber security against the business needs [5].

### 2.2. Models of Cyber Security

Traditional cyber-attack taxonomies such as the infamous McCumber cube model (1991) [6], have always proven to be successful in the past, with its model being used in businesses to detect traditional cyber-attacks, and being able to counterattack using traditional counter measures. The McCumber model was the first to formally evaluate information security, merging theory with practical implementations in policy, education, and technology. It allowed businesses to follow a model based on the Software Development Life Cycle (SDLC) bringing together its own states of security services, security counter measures and information states. The McCumber Cube looks at how data is stored, transmitted and processed. It also looks at data being confidential, having integrity and being available, and is inclusive of data having states of non-repudiation and authentication. Lastly, people, policy and practices plus technology have a direct impact on how data is used and kept safely. The McCumber cube model is still strong regarding how data is handled; however, modern-day taxonomies are now playing an even more key role in the detection and prevention of cyber-attacks. Newer elements as explained in [6] such as "time" are now added to the original elements of people, processes, and technology to help secure a system from modern-day cyber-attacks. By introducing the notion of "time" to the cube, it allows for a greater expansion in understanding how these elements of people, processes, and technology, with the fourth element being "time," are able to help secure a system. Whilst the McCumber cube uses basic attack taxonomy dimensions such as Confidentiality, Integrity and Availability (CIA) as basic elements to analyze an attack, the fourth element of "time" contributes to how these cyber-attacks can be detected and prevented at an earlier stage of its life. When evaluating an attack, the element of "time" has a direct impact on these dimensions of CIA, in terms of authentication, authorization and accountability. This gives an example of a cyber-attack progressing through the element of "time", attacking how and where data lives at any one point in our technology setup especially when connected to the internet. Here, "time" can easily destroy data if not acted upon quick enough from the cyber-attacks and threats. Countermeasures become an important part of the data cycle if not acted upon fast enough to stop the attack. The addition of "time" into the McCumber cube model is recognition that cyber-attacks do progress and will change the way data is handled to reflect the current state of the attack.

With data constantly increasing with the creation of Big Data [7], internet traffic has become harder to control and manage; therefore, a change in working methods needs to be adopted. A more intelligent way of working needs to be implemented towards newer models of smart roads, highways and living. The ideology of anything "Smart" leads to the internet having to work with newer technologies such as artificial intelligence and machine learning in order to be able to detect cyber-attacks on a platform, which is quite different from traditional ecosystems of the world wide web [8]. The difference here lies in the web,

of understanding how technology has moved on, not only from a cyber-security point of view but also from the view of the cyber-criminal activities that receive funding on the dark web, which enable these groups to advance in technologies themselves to gain power [9]. The use of machine learning in the detection of criminal activities online, and its ability to recognize any intrusions happening or about to happen, is one that is quickly catching on with businesses. The flipside of this is that the cyber-criminals have also gained strength in technology and they themselves are using intelligent ways to drive the dark web into larger criminal activities online. An attacker's perspective is one that is important to understand, as this can give further understanding to the security mindset that is required to combat the offence and increase the defense through the various methods discussed in this paper. With technology evolving, intelligent software is more pertinent than standard methods of trying to "reduce and increase" crime at the same time on the internet, leading to more possible cyber-attacks [9].

Whilst intelligent technology using machine learning is growing in number and trust, understanding the uptake of this software and how it is emerging is attracting curiosity. Whilst plenty of research and testing is being conducted within the government, as well as academic and larger organizations, it is important to understand how small–to-medium enterprises (SMEs) are able to cope and handle the fast-paced activities happening online. In particular, it is important to understand how developed nations such as Wales are able to use these technologies, and how their awareness and willingness to explore these concepts are used in practice when protecting their data.

The next sections, which include the literature review and methodology, will take a further look at how developing nations are coping and how comparisons are made of intelligent software uses and how awareness within the internet and networking infrastructure is managed. These next sections will also discuss the datasets and models for machine learning algorithms that make a difference when added to traditional devices in pointing out attacks close to zero-day and beyond, plus new adversarial effects the training data might have on how intelligent softwares can be used to move forward.

## 3. Literature Review

### 3.1. Intelligent Software Uptake in SMEs

Data are the new gold and hold high value in this day and age; this cannot be afforded to be stolen or frauded in any which way for businesses, or they will face financial and reputable loss. As data travel rapidly over the internet, the surface area where data is passing through can get rough, as this terrain, if not protected, can invite cyber threats to cause chaos. Traditional software and techniques do not perform well any more in the detection of malware, intrusion, spam and phishing, together with IP traffic classifications. This is where the adoption of intelligent software that uses machine learning algorithms could be the answer in the prevention of attacks on the internet. Soni and Bhushan's paper recognized, " … *how several algorithms of Machine Learning can be used to overcome the popular issues faced by cyber security*." Their study was particularly relevant in that machine learning cleverly monitored intrusion, detection, and was able to then offer the protection of internet traffic based on this notion of "time" [10]. Vakakis, N. et al. (2019) also agreed with Soni and Bushan in that older models of intelligent software were also wearing thin on their success stories and considered the use of newer anomalous models to overcome this cyber challenge in maintaining ambitious standards of cyber security [11]. Testing and implementing these anomaly detection methodologies, using the right datasets, showed better accuracy for pattern recognition of cyber threats, especially within smart devices and IoTs, and other application that apply in smart environments similar to SMEs. Wylde, V. et al. (2021) explored machine learning-based intrusion, detection, and prevention software becoming a priority in the realization of keeping SME data secure and safe with the integration of real-world objects and IoT, with understanding how these machine learning techniques and artificial intelligence can help secure zero-day attacks [12]. Rawindaran et al. took particular interest in the SME market and showcased an experimental scenario in which

the intrusion, detection and prevention models were compared, and the views of the SME were examined. The study looked at the various approaches in identifying ways to detect and protect any intrusions coming into the network and what operating devices would help in this process [13].

*3.2. Intelligent Software Uptake in SMEs from Developed vs. Developing Countries*

The developing country of South Africa saw the first example of how cyber security practices were followed by SMEs through a qualitative approach. With developing countries now accounting for the majority of users, 2.5 billion users online, this provides the need to understand how these users are protecting their data whilst navigating the internet. Internet penetration has hugely different dynamics in developed countries, where it is at a level of 81%, compared to 40% in developing countries and 15% in the least-developed countries. Whilst the 40% penetration rate of developing countries is far lower than that of the developed countries, the targeted audience using the internet was much higher. This audience comes from those in the developing countries taking advantage of the right internet traffic to form international relationships through new supply chains to increase sales. This "*sales activity*" was done through increased communication across the internet, thus reducing costs and an increase in quicker transaction processing [14]. Kabanda et al. (2018) explored all the positives of using the internet for these financial benefits; the cyber threats naturally increased with the increase of transactions and the cyber landscape [15]. Von Solms and Kritzinger (2011) stated that SMEs in South Africa had " . . . *several negative comments about Africa's lack of preparedness to combat cybercrime*" [16] whilst Dlamini and Modise (2013) added that: " . . . *South Africa remains one of top three countries that are targeted by phishing attacks*" [17].

Rawindaran et al. (2021) conducted similar research with developed nations, on a pilot study of SMEs within Wales, during the COVID lockdown period of March 2020. Rawindaran's paper took examples of previous research from other developed nations such as Spain and Australia as benchmarks to find similar behavior in SME businesses and common grounds in how people largely contributed to the effects of cyber threats and security. Rawindaran's study revealed gaps from the human element of awareness and readiness to tackle cybercrime through the education of using intelligent software. Rawindaran's pilot study further confirmed the barriers and reasons for the low adoption of intelligent techniques through survey questionnaires as methods of extracting working information from these SMEs. Findings from Rawindaran's study showed from the results that SMEs relied on the expertise of their IT team to set an example on how the ideas of advanced software can be used to move forward in the further protection of their data. Equally, management wanted to learn and understand more about cyber security and the effects it could have on their business. The exposure to terminologies such as artificial intelligence and machine learning were extremely low, in that some SMEs were not interested and did not think it was applicable to their industry in trying to protect their data. The awareness of the cyber security pool of participants was very weak and suggestions were made for further education and training in the sector. Rawindaran's paper also highlighted the importance of government providing funding through grants, subsidies, and financial assistance to help with the cost of implementation in getting the right technology and the right expertise involved [18].

Mutalib, M.M.A., et al. (2021), focused on how SMEs coped with cyber-attacks in a developing country such as Malaysia, compared to the UK. Mutalib confirmed that SMEs often found it difficult when experiencing a cyber-attack on how to recover their business in the aftermath of the tragedy. SMEs in Mutalib's study were found to have had no contingency planning, no preparation, no expertise nor experiences in dealing with a cyber-attack. Mutalib's research proposed a new framework to help these SMEs find strategic ways in which to assess their situational risk and plan for this contingency, as well as suggested remedial actions. Mutalib's study used a methodology to investigate SME cyber security behavior and challenges based on ideas of "how many" and "how effectively"

SME users reacted to malware attacks. Mutalib utilized a platform called the Coordinated Malware Eradication and Remediation Platform (CMERP) as a base framework for malware data collection and statistical analysis. This framework also used the National Institute of Standards and Technology (NIST) cyber security standards and policies for SMEs to improve their cyber security efforts by implementing NIST incident response life cycle and gather outputs [19].

Alahmari and Duncan (2021) highlighted that SMEs were constantly subjected to potential barriers in their cyber security risk management approaches within the business. The study used a semi-structured interview process with twenty decision-makers in Saudi's SMEs landscape using a qualitative exploratory approach. The result from this study showed barriers which played a significant role in preventing SME's decision-makers from investing further in cybersecurity risk management to mitigate this risk within the business. Alahmari and Duncan's main focus was understanding these barriers and how cybersecurity risks could be managed to minimize losses by investing in cyber security within SME businesses. Alahmari and Duncan explained that SMEs in existing businesses in both developed and developing countries were suffering losses and attacks in excess of 90% [20].

Various other studies included that of Kalhoro et al. (2021) and Emer, A. (2021). Kalhoro suggested that practicing "cyber hygiene" within the SME industry would provide a better protection, better security, monitoring, and maintenance of the networks of software development organizations. Raising the knowledge and awareness of cyber hygiene amongst software engineers through employee training programs was employed in this study's approach [21]. Likewise, the "hygiene" factor was also discussed by Emer who went on to give an introduction of an assessment tool that allowed for a check list to cover all aspects of the business within a smaller SME environment. The study was conducted across the developed and developing nations of Italy, Ecuador, and Thailand [22].

Cyber security concepts often start with the realization of technology and how technology dependencies create a safe haven for data. Being resilient in cyber security offers a benchmark for working towards the various sections of the McCumber cube model [1], bringing compliance to SMEs in the form of guidelines and with a refocus on policy, standards, and rules within a business.

## 4. Methodology

The methodology used in this paper follows positivism in the form of a quantitative approach using a survey questionnaire. It involves distribution of the survey to SMEs in Wales using convenience sampling in which data was collected from 122 sampled respondents. Qualtrics was used to create the survey and distribute it [23]. A convenience sample was used through social media platforms such as LinkedIn and Twitter and through email distribution using a snowball strategy through the researchers' professional networks and requests to forward the invitation on to other Welsh SMEs. The research adopted this sampling approach for two reasons, with social media links being the most widely used social networking application amongst SMES in Wales. Second, a convenient sampling strategy enables advertising of the study to spread through a network of Welsh SMEs [24]. Potential respondents who showed initial interest in the survey were directed to the survey platform to read the information about the purpose of the study and the background of the research team. Respondents had to tick the consent box and start the survey. All survey answers were anonymous. The survey was a time-based frame and given the timeline of two months. According to Greenfield (2002) [25], a timeframe is important for larger projects in collection of a larger sample pot where analysis and evaluation can be further examined. The participant feedback was anonymized unless the participant wished to be named and contacted for further research.

*Design of Questionnaire*

A pilot study was conducted in Wales across 30 SMEs as an initial study. There was a set of 34 questions asked from participants. Statistical analysis was used and the differences

between various experimental groups were analyzed with a *p*-value and Chi-squared test. All analyses including Chi-squared tests were performed using Excel software. Any value of $p \leq 0.05$ (two-sided) was considered significant [26].

The survey question was based on the six hypothesis testing questions as shown below:

| |
|---|
| Q1: Does being a Decision Maker impact the awareness of machine learning cyber security (MLCS) and the chosen software package adopted within the SME business? |
| Q2: Does having a specific Role within the SME business impact the awareness of MLCS and the chosen software package adopted? |
| Q3: Does Gender play a part in awareness of MLCS, and the chosen software package adopted? |
| Q4: Does Age play a part in awareness of MLCS, and the chosen software package adopted? |
| Q5: Does being Educated within the SME business impact the awareness of MLCS and the chosen software package adopted within the SME business? |
| Q6: Does the Size of the SME impact the awareness of MLCS, and the chosen software package adopted within the SME business? |

The questionnaire focused on collecting information on respondents' roles and personal information within the organization as well as management, technical, and non-technical expertise. The questions focused on the individual's age range, identity, and their role, education, professional certifications, and industry. Each question was then compared to a question on SME awareness on the cyber security software package used within the business showing the participants cyber security awareness. For each question compared to this awareness, the data collected were cleaned by removing blanks to reveal final lists that contained data with integrity. Once cleaned, the observed data were processed through statistical analysis using the Chi-squared test to find the *p*-values.

The survey summarized the opinions on how SMEs could raise awareness of machine learning cyber security technology and how this could be made better, and if it were appropriate to follow up on the responses moving further into the research in the future. The survey targeted varied sizes of SMEs, micro, small, or medium, based on the number of people in the company. The SMEs also got an opportunity to answer the relevant questions pertaining to their understanding and awareness of intelligent software and their current cyber support packages. The ethics process, approved by the Cardiff School of Technologies Ethics Committee, was followed throughout this study. Informed consent was taken from all participants. No personal data were collected, and all the data were anonymized.

## 5. Results and Analysis

The study's results and analysis are identified and analyzed in this section. The education level was divided into school education right through to university and professional certification levels. The majority of the population sampled had a Bachelor's or Master's degree as shown in Figure 1 below.

Figure 1 shows the varied education levels identified as part of the questions asked. The results showed that the majority of respondents had a Master's degree education level coupled with a professional commercial certification. This combination was quite common in respondents running their business and employing trained people as part of the team; however, the percentage of respondents having a professional certification on top of their education was exceptionally low. The professional certifications included project management skills such as SCRUM and PRINCE2, financial certifications such as FCCA and some Cisco software certifications, to name a variety within these SME industries.

Figure 2 evaluates the age and identity of the respondents who participated in the study, with a large majority being between the ages of 36–45 years of age, at 39%, followed by 46–55, with 27%, with an almost balance gender measure of men and women.

Data on the types of industry these SMEs covered were also captured to understand the backgrounds and the experiences of different SME industries impacting the cyber security knowledge.

Figure 3 below shows that the respondents covered a wide variety of industries, both horizontal and vertical.

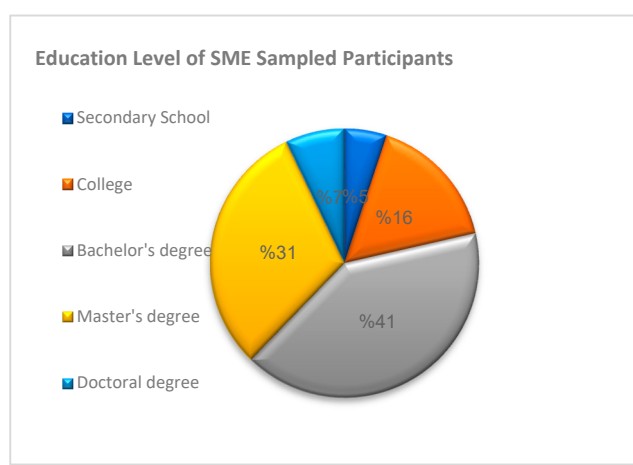

**Figure 1.** Education and Professional Certifications.

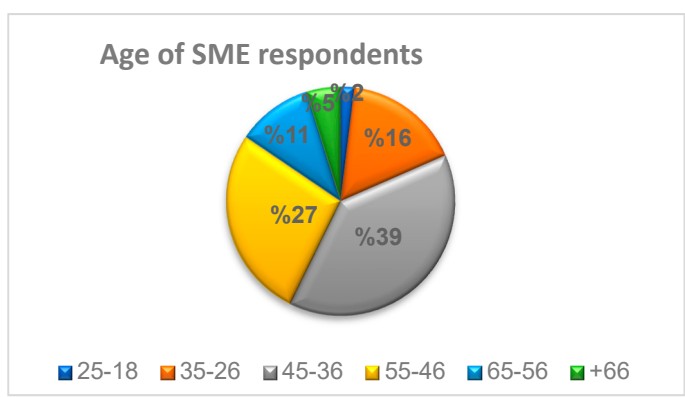

**Figure 2.** Age of Respondents.

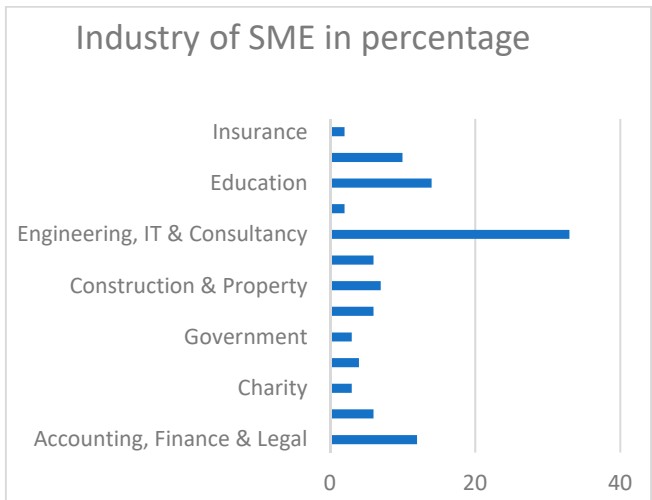

**Figure 3.** Industry of SME.

Figure 3 shows industries ranging from accounting, finance and legal, construction, manufacturing, and engineering/IT services. More niche industries such as charity, logistics and hospitality were also identified. The diversity of industries including those of government and council also saw respondents from those sectors.

Figure 4 below shows the size of the SME industry within these sectors. Just under half of the participant SMEs were in the category of small SMEs with 0–10 employees, at 47%, with 11% ranging from 11–20 employees. Employee numbers of 21–30 were 7% of the respondents with the rest, 33%, covering employee counts of above 50.

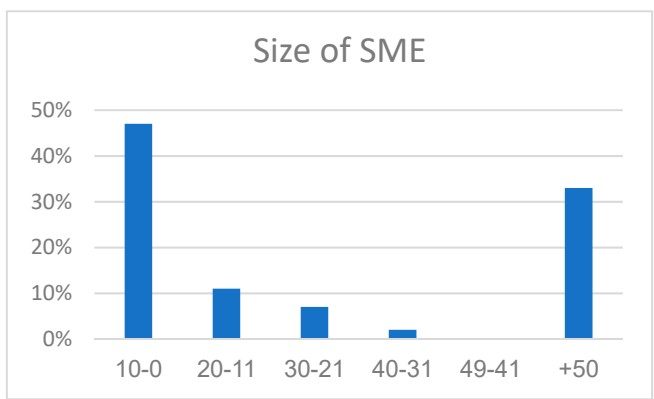

**Figure 4.** Size of SME participants.

Figure 4 above shows the various size measures of SME categories depending on the types of industry and how the businesses are managed, as well as the financial turnover.

Figure 5 below gives a snapshot of the types of respondents surveyed. Majority of the respondents were from a management background, followed by a split of technical managers and the technical team.

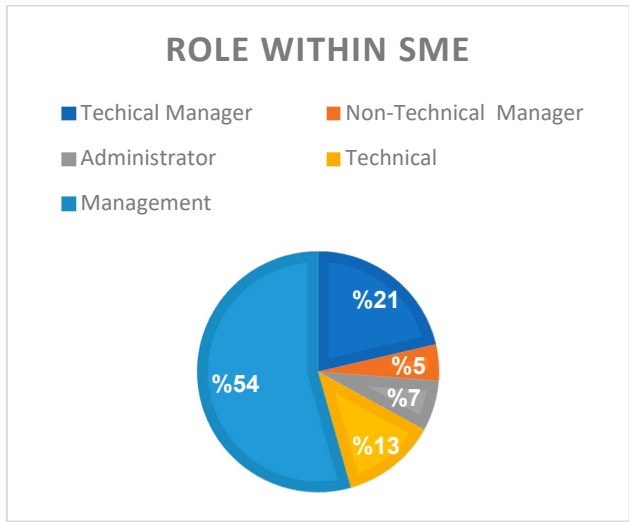

**Figure 5.** Role within SMEs.

There was also a mix of non-technical managers coupled with a few administrators answering the survey to the best of their ability and knowledge. Figure 6 below completes the section by identifying the decision makers within this process and the influencers with the SME.

Figure 6 identifies that 85% were decision makers and influencers of the SME and just under 15% had no say within the decision-making aspects of the business but were users of the systems.

All six sections above highlight the various variables that became pertinent to this study in terms of the SMEs being surveyed. The next section goes on to see the results of the machine learning knowledge of these SMEs within the cyber security aspects of the business.

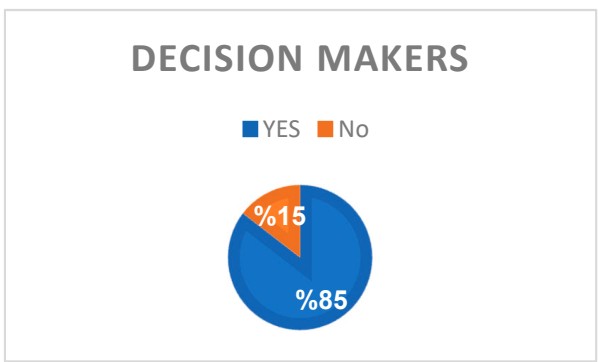

**Figure 6.** Decision Makers within SMEs.

*5.1. Knowledge and Awareness of Machine Learning in Cyber Security*

This section focuses on asking the SMEs of their knowledge and awareness of ML in cyber security software packages to protect their business from cyber threats, and if so, to state the software, and if no, then to give a reason. This question was pertinent as it then led us to identify how the SMEs detect cyber-attacks and if ML was used as part of the options to secure their network and data.

The results from Figure 7 began with the awareness of MLCS, with the majority of 73% saying "YES" to being aware of MLCS. Respondents who were unaware comprised 10% whist 8% confirmed an answer of "NO". A total of 9% of these respondents explained why they had said "NO or Unaware".

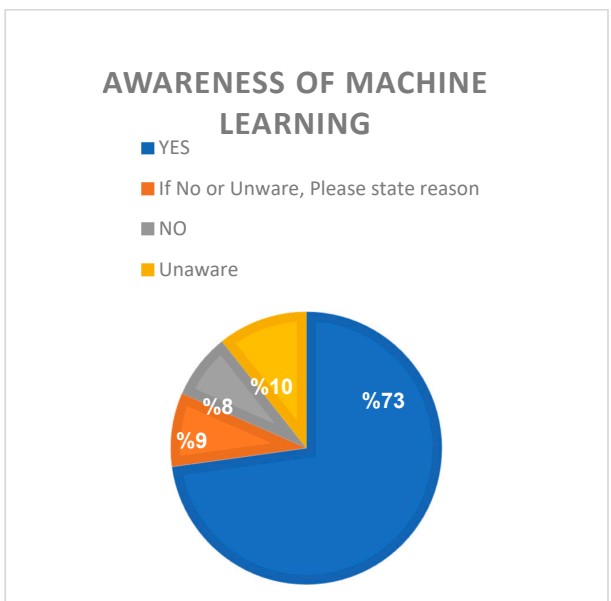

**Figure 7.** Awareness of Machine Learning.

The respondents who gave these explanations identified that the support from their IT team was enough for them to not need to understand machine learning and relied on those companies to look after the SME needs, whilst others said that they were not sure if their software license had expired, only used simple anti-virus, or were run by third-party organizations such as a university. The above questions led to the SME being asked of the challenges of using machine learning in cyber security packages as shown in Figure 8. Nearly 72% stated that it was due to just being unaware, 10% stated that these challenges were due to barriers to technical expertise whilst 8% said it was due to the excessive cost of implementation and use.

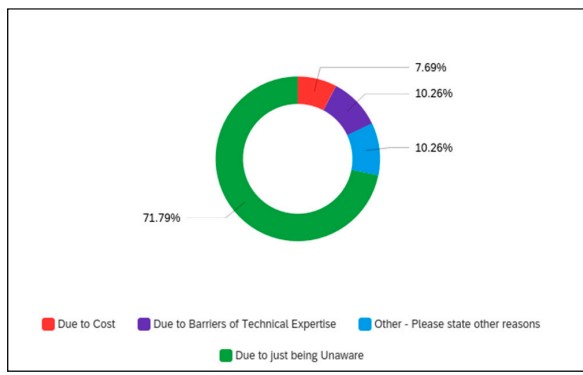

**Figure 8.** Challenges to implementing machine learning.

The last 10% surveyed came back with results identifying that the SME head office had dealt with any cyber security issues and that the IT company supported the SME using this technology. The challenges in Figure 8 above were made clear by the respondents in their answers to the survey. Figure 9 below dug deeper, looking at the awareness of the algorithms of machine learning that were used in these SME cyber security packages and if these SMEs were aware of them. From the 73% of those that said "YES" from Figure 7, approximately 70% of those respondents were aware of the algorithms and recognized key words such as Neural Networks, Deep Learning, Support Vector Mechanism and so on.

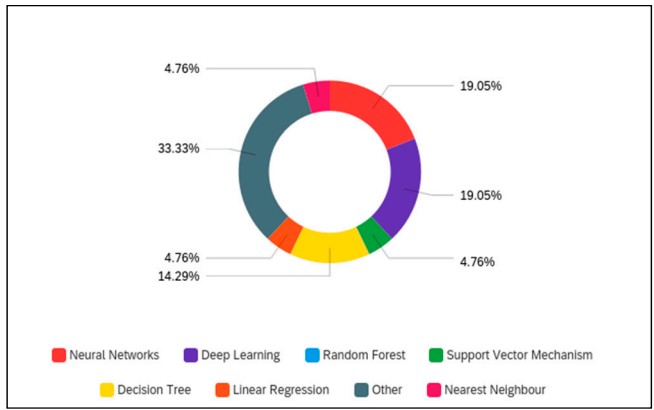

**Figure 9.** Awareness of Machine Learning Algorithms.

The remaining 30% from the respondents stated that whilst they were aware of machine learning they were not aware of the algorithm or unsure. Some mentioned that it was unknown to them, and that this data were confidential, whilst a few mentioned that their IT support company knew. The same pool of participants here added a comment to note that the implementation of CS software packages within their SME business started from 2018 onwards.

### 5.2. Statistical Hypothesis Testing

*p*-Values are measurements used to quantify the statistical significance of any observed results [26]. A *p*-Value is always based two hypotheses. For the first one, being a null hypothesis, H0, is normally assumed that there is no difference or has no effect of a treatment or an exposure. The second is an alternative hypothesis, H1, and is often an assumption that the null hypothesis is untrue.

From the results returned, Table 1 shows the questions asked to verify the null/alternative hypotheses produced prior to the collection of results in the survey questionnaire. The table below produced six null hypothesis and six alternative hypotheses based on the elements questioned in the survey.

**Table 1.** Null/Alternative Hypotheses.

| Question | Null Hypothesis | Alternative Hypothesis |
|---|---|---|
| Q1: Does being a Decision Maker impact the awareness of MLCS and the chosen software package adopted within the SME business? | H0: Being a Decision Maker has NO IMPACT in awareness of MLCS and the chosen software package adopted within the SME business | H1: Being a Decision Maker has an IMPACT in awareness of MLCS and the chosen software package adopted within the SME business |
| Q2: Does having a specific Role within the SME business impact the awareness of MLCS and the chosen software package adopted? | H0: Having a specific Role within the SME business has NO IMPACT the awareness of MLCS and the chosen software package adopted | H1: Having a specific Role within the SME business has an IMPACT the awareness of MLCS and the chosen software package adopted |
| Q3: Does Gender play a part in awareness of MLCS, and the chosen software package adopted? | H0: Gender has NO IMPACT on awareness of MLCS and the chosen software package adopted | H1: Gender has an IMPACT on awareness of MLCS and the chosen software package adopted |
| Q4: Does Age play a part in awareness of MLCS, and the chosen software package adopted? | H0: Age has NO IMPACT on awareness of MLCS and the chosen software package adopted | H1: Age has an IMPACT on awareness of MLCS and the chosen software package adopted |
| Q5: Does being Educated within the SME business impact the awareness of MLCS and the chosen software package adopted within the SME business? | H0: Being Educated has NO IMPACT on the awareness of MLCS and the chosen software package adopted within the SME business | H1: Being Educated has IMPACT on the awareness of MLCS and the chosen software package adopted within the SME business |
| Q6: Does the Size of the SME impact the awareness of MLCS, and the chosen software package adopted within the SME business? | H0: The Size of the SME has NO IMPACT on the awareness of MLCS and the chosen software package adopted within the SME business | H1: The Size of the SME has an IMPACT on the awareness of MLCS and the chosen software package adopted within the SME business |

Table 2 below, shows the results of the statistical analysis performed on the six attributes listed, giving the below *p*-Values. The results are displayed in the table below.

**Table 2.** Attributes and their calculated *p*-Values.

| Attributes | *p*-Value | H0/H1 |
|---|---|---|
| Education | 0.03 | H1 |
| Age | 0.4 | H0 |
| Gender | 0.6 | H0 |
| Size of SME | 0.005 | H1 |
| Role | 0.8 | H0 |
| Decision Maker | 0.96 | H0 |

Table 2 above shows that if the *p*-Value is greater than 0.05, then the result is not statistically significant. A *p*-Value less than 0.05 (typically $\leq 0.05$) is statistically significant. This states the hypothesis is rejected and to accept the alternative hypothesis. A *p*-Value higher than 0.05 (>0.05) is not statistically significant.

Using the *p*-Value measurement of significance, the elements were compared in terms of the affects they had towards the research on being aware of MLCS. The below Figure 10 shows the elements that made an impact on the research and had a *p*-Value less than 0.05. The figure also shows the elements that were non-impactful in the results. The six main

elements that were measured were Role, Decision Maker, Size of the SME, Gender, Age, and Education of the participants surveyed within the SME pool.

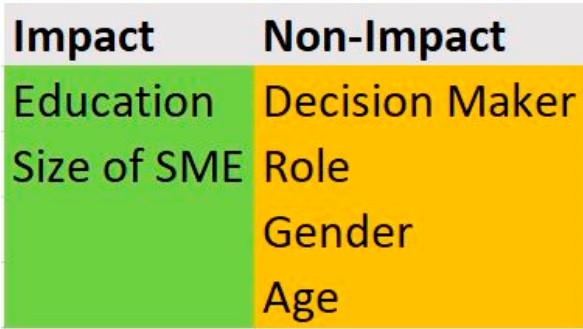

**Figure 10.** *p*-Value impact of variables.

The results concluded that two elements had a significant impact on how the awareness of MLCS can be raised within an SME environment and its structure. These elements were Education and Size of the SMEs. Elements that had no relationship to this hypothesis were the participant's role within the SME business and if they were decision makers or not. Gender and age also showed no impactful relationship in raising awareness of cyber security within the SME business. It was apparent that by having the right educated people in place within the SMEs, this made the relationship of awareness to MLCS software packages more effective in protecting their data, and gave their business a more informed choice within the SME. The results above in Figure 10 have proven that the smaller the SME and the more educated the people who work within this environment, this contributes to raising awareness of MLCS moving forward in making better choices, in terms of getting the right technology in place to keep their data safe.

## 6. Limitations

Limitations from this study included that a large sample of participants could have increased the detailed data collected, formulating a more in-depth analysis of the outcomes and results. Understanding further relationships of these elements discovered and how these elements made an impact on the findings could have resulted in a broader framework being used to look at the patterns of answers from this wider sample pool of participants. Further studying of SMEs' cybersecurity decision making philosophy could minimize potential risks, pushing for the adoption of more useful technologies to compliment the study, and those SMEs looking for financial backing to help bridge the gap in knowledge and awareness. Notably, more exploration of the studies of cybersecurity risk management in SMEs is needed, expanding on data from more developed and developing countries, creating a larger sample across varying countries. Similar to other studies, this paper has limitations that need to be acknowledged. Once the limitation of the sample size can be overcome, the collected results could potentially show a variety in results. The COVID pandemic may have changed some answers and behavior of decision makers, giving perhaps a more unprecedented answers during the pandemic. This could be due to various reasons of economic, social, and political changes occurring around the world. Additionally, having a variety in geographic scope increases the reach, enhancing understanding of the global impact of these questions on SMEs worldwide. Limitations in this study offer the wider scale of trying to understand different risk management standards and technologies currently being used, and if a certain method is already helping to reduce cyber risks in SMEs significantly. The objective of the study was to identify awareness of intelligent software within Welsh SMEs and to investigate the potential elements that could contribute to this current gap. Future studies will focus on exploring these relationships further in nature and adoptability. These relationships could open insights into developing more accurate solutions theoretically and practically. Thus, future research can also replicate the

present study by extending the research sample or including another geographic scope focusing on SMEs in industry.

## 7. Conclusions

Intelligent software is proving its popularity and power in offering the defense required to combat the offence from cyber-attacks. Attacks made online are becoming more intelligent in nature, utilizing artificial intelligence and machine learning algorithms to create havoc, either to steal data or claim power in stealing the data. This study highlights the vulnerability of Welsh SMEs and their uptake on machine learning cyber security (MLCS) software packages and why there is a lack of uptake within the SME community. The study also focuses on Welsh SMEs cybersecurity awareness, knowledge of cybersecurity, and how they contribute to towards SMEs' resistance or willingness to move forward with this concept. It also highlighted how engagement with cybersecurity agencies can boost SMEs' digital confidence in tackling complex problems, however this is not seen as important within the SME sector due to their lack of digital confidence in the area of cyber security. SMEs need to be aware that cyberattacks will eventually hit their business and that it is only a matter of time, and if they are not prepared for this process, the loss will be detrimental to the SME business. Getting SMEs to understand this concept is indeed a challenge in itself and can only get better if digital maturity takes place. The objective of this research was to quantify and substantiate the various different cyber security elements of people, processes, and technology. In particular, the dependency of the human elements on the effects these elements had on cyber-attacks and the security that comes with it were analyzed. The research was to explore which areas of the human element needed addressing in order to understand why SMEs just do not use the right software to help them keep their data safe. Government laws and schemes and programs are made available to the general public and to SMES, however, the uptake is very low within the SMEs in Wales. It was evident in the results of the research that the element of "education" within a Welsh SME and the element of "size of this SME" played a huge part in understanding SMEs' "awareness" of how intelligent technology can be used to keep their data safe. Making the right choices in as to which combination of technology SMEs should use and the assorted options on how machine learning and artificial intelligence operates can be utilized to increase the measures of ongoing cyber security to combat cyber-criminal activities. Those SMEs who had a more educated team of people working together proved to have a better understanding and have awareness of the cyber security technologies being used and deployed, alongside the capabilities of machine learning implementations within the technology. This research showed that the more aware SMEs were on intelligent software, the more the SME would adapt the right model in place to protect their business. The size of SMEs played a crucial part in the significance of the findings in the research proving that the smaller the business of educated people running it, the safer the environment and the increased SME awareness of the intelligent software. As the SME size grew, the awareness became diluted until the SME organizations could afford to build a more strategic team to manage various components within the business to then fall into the larger business category. This larger business would then have the capacity and the money to be able to run test environments to help trigger and control cyber-attacks using more complex technology to control these attacks. The literature within this research revealed how SME risks of cyber-attacks were identified and how a balance was required to run and finance the business whilst trying to implement the right IT solution for the SME business. The literature also strengthens this notion that funding, resourcing and awareness all play a big part in keeping these attacks at bay. The research also went on to cover the "full circle" of what SMEs in Wales need in order to protect their business, and through the path of education and getting the right people into the business, we can help build this awareness and maintain a healthy growth of the SME business without compromising on the cyber security aspects of the business. Elements such as age, being a decision maker, having a specific role, or being male or female did not have any contribution or impact on the awareness on cyber security,

causing a neutral non-dependency bias in terms of recruitment and growth of the business. As long as SMEs are educated and work within a smaller-sized environment, this research proved that these two elements helped increase the awareness of cyber security. Having the right education levels and the right number of people within the SME, from this research was enough to protect data and continue running their business safely and securely.

The research focused on the changes set to challenge SMEs in how they deal with cyber-attacks and threats that are constantly reoccurring as long as SMEs are connected to the internet. The research and state-of-the-art literature review will expand the views and contributions of research to how SMEs are running their business on a knife's edge without the realization that an attack is imminent and could cause chaos and extreme financial loss in the process. The research explored the adoption of intelligent software and the current support structure in place for SMEs who lack awareness in this area and the guidance and help from government and governance has at ground level that bypasses many SMEs getting the desired outcome required to keep their business safe.

The motivation of this research stemmed from the SMEs being targeted by attacks daily and what was being done about it. SMEs concerned over practical problems have initiated this research in trying to find the best solution moving forward to reduce the crimes occurring on the internet directly impacting the SMEs business and their data in Wales. The specific urgency imposed by this challenging aim formed the motivation behind this research.

**Author Contributions:** Conceptualization, N.R., A.J. and E.P.; Methodology, N.R. and A.J.; Software, N.R.; Investigation, N.R.; Resources, E.P.; Supervision, A.J.; Funding acquisition, A.J. All authors have read and agreed to the published version of the manuscript.

**Funding:** This paper has been supported by the KESS2, Knowledge Economy Skills Scholarships, and Aytel Systems Ltd., Cardiff, UK and Cardiff School of Technologies—Cardiff Metropolitan University.

**Informed Consent Statement:** Not applicable.

**Data Availability Statement:** All data has been present in main text.

**Acknowledgments:** This paper has been supported by Cyber Resilience Wales.

**Conflicts of Interest:** The authors declare no conflict of interest.

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
