# Peer review of "Exploration of the Impact of Cybersecurity Awareness on Small and Medium Enterprises (SMEs) in Wales Using Intelligent Software to Combat Cybercrime"

_computers, doi:10.3390/computers11120174_

Round 1
Reviewer 1 Report
The paper makes a good contribution to the literature. However, some changes are necessary before the work can be published. I hope my comments are seen as being constructive.
Title: Please remove "October 2022" from the title and include Wales (see comments on introduction). Please also delete the reference to a quantitative study. This is because no statistical model is presented - refer to papers on positivist research in the Information Systems domain, which also uses questionnaires.
Introduction: As the journal is not specific to cyber security, the authors should consider including a short description of cyber-specific terminologies used in the paper (e.g., McCumber cube model, CIA, data-at-rest, data-in-transit, data-in-process). Having a reference is good, and this should be included as well. With reference to the survey the authors mention both Wales and UK? As the authors would know, Wales is part of UK. If other regions from England have been survey, then they need to state Wales and England (both being parts of UK). However, considering that the lead author is from Cardiff, there is a possibility that the survey was done only in Wales. If so, then please say so.
Literature review: The section needs further work. Note that the first two paragraphs all refer to the same reference, which has been cited four times. It appears that the authors are paraphrasing the cited work? This is also true for para 3 of the literature review and which focuses only on the work of Vakakis et al (2019); Section 2.2 focus is on Kabanda et al. (2018), and so on... The objective of the literature review is literature synthesis, for which the authors must demonstrate that they have a good grasp of the literature. I also suggest that the literature review be shortened.
Methodology: I think the authors should say that the survey was conducted in Wales, rather than trying to generalise to all of the UK and other developed countries. The authors will need to reduce content on sampling. I think the authors' approach is that of Convenience Sampling. I would include a short write-up on this approach rather than try to project the study results to be generalisable.
New section needed: Please add a short section on the design of the questionnaire (after methodology and before results). Did the authors conduct pilot testing of the questionnaire?
Results and Analysis: Data shown in Figures 3-11 can be included in tables or the general write-up. For example, men (58%) and the rest are women. Why is it necessary to have a pie chart here?
Statistical Hypothesis Testing: The authors explain basic concepts. I would reduce the content and make this section a sub-section to results and analysis. The claim that this study is quantitative is based on the authors having done hypothesis testing. However, there is no statistical model with constructs being developed, and this should come from the literature. Hypothesis testing appears to be an afterthought. The authors may like to read papers on positivist research in the Information Systems domain, which also uses questionnaires, to consult the design of such studies for a future publication!
Minor:
(1) Lines 51-52: Include the year for Easttom and Butler reference.
(2) Line 73: " middle market of SMEs"?
(3) Figure 1: The caption reads "Authors interpretation of Ben-David et al. five “Core Forces”. Does "Authors" refer to Kabanda et al. (2018) or the authors of the paper. Please check the paragraph just before the figure. Also note that for Ben-David et al. the year is missing. This needs to be checked for all references.
(4) Reference formatting: I am not sure why authors are using both numbered references [X][Y] and also (author, year)?
Author Response
Title: This has now been changed to reflect the content and aim of the paper
Introduction: This section now has Wales as its focus point and centre of the study. Explanations of technical cyber terms are simplified and referenced.
Literature Review: This section has now been shorted and hopefully discussed with a better flow and a synthesis to the reading and connection to the theme of SMEs cyber works in keeping their data safe.
Methodology: Thank you very much for your guidance here of Convenience Sampling. I have hopefully added a fresh version to this section to explain the train of thought in this paper.
New section: Design of survey is now added based on hypothesis testing.
Results: have been amended as guided
Statistical Hypothesis Testing: Reduced and integrated into smaller section
Minor:
(1) Lines 51-52: Include the year for Easttom and Butler reference. ( Removed)
(2) Line 73: " middle market of SMEs"? ( Corrected)
(3) Figure 1: The caption reads "Authors interpretation of Ben-David et al. five “Core Forces”. Does "Authors" refer to Kabanda et al. (2018) or the authors of the paper. Please check the paragraph just before the figure. Also note that for Ben-David et al. the year is missing. This needs to be checked for all references. ( Removed)
(4) Reference formatting: I am not sure why authors are using both numbered references [X][Y] and also (author, year)? ( Hopefully I have corrected this format. )
Once again thank you for your esteemed advice. Greatly appreciated.
Reviewer 2 Report
The paper presents a quantitative study in relation to SME awareness of ML cyber security software packages.
It is not clear what the relationship between the McCumber cube model and the addition of "time" is related to MLCS and the quantitative study conducted in this paper. The introduction talks about this at length and it is referred to in 2.1, but from 2.2 onwards it moves on to other things. How do the questions in the questionnaire specifically address this? There is a disjoint between the questions and the literature review. There need to be explanations linking these.
Also, it is not clear why SMEs need to know the details of ML algorithms. What is the problem with SMEs treating ML solutions as black boxes that they purchase as part of a security software package?
There should be a clear statement regarding the contribution of this paper.
While the paper presents problems regarding awareness of MLCS in SMEs, the paper does not propose any solutions to this problem. It would be good to have a section discussing potential solutions.
Minor comments:
- Acronym definitions are in lowercase in the abstract but at other times, the first letters are in uppercase.
- MCLS used at times instead of MLCS.
Author Response
I have changed the flow and content to read better , hoping that this will be good to the reader to understand the story better reflecting the short comings of the uptake of newer technologies for SMEs in Wales.
Minor comments:
- Acronym definitions are in lowercase in the abstract but at other times, the first letters are in uppercase. ( Now Changed )
- MCLS used at times instead of MLCS. ( all corrected )
Thank you very much for your comments and guidance here.
Round 2
Reviewer 2 Report
After the revision, the paper is now smoother and more focused. I'm satisfied with the changes.